# Clinical and survival differences during separate COVID-19 surges: Investigating the impact of the Sars-CoV-2 alpha variant in critical care patients

**Andrew I. Ritchie[1,2◉], Owais Kadwani[2◉], Dina Saleh[2], Behrad Baharlo[2], Lesley R. Broomhead[2], Paul Randell[2], Umeer Waheed[2], Maie Templeton[2], Elizabeth Brown[2], Richard Stümpfle[2], Parind Patel[2], Stephen J. Brett[2,3◉], Sanooj Soni[2,3◉]***

**1** National Heart and Lung Institute, Imperial College, London, United Kingdom, **2** Department of Critical Care, Imperial College Healthcare NHS Trust, London, United Kingdom, **3** Division of Anaesthetics, Pain Medicine and Intensive Care, Imperial College, London, United Kingdom

◉ These authors contributed equally to this work.
* s.soni@imperial.ac.uk

## Abstract

A number of studies have highlighted physiological data from the first surge in critically unwell Covid-19 patients but there is a paucity of data describing emerging variants of SARS-CoV-2, such as B.1.1.7. We compared ventilatory parameters, biochemical and physiological data and mortality between the first and second COVID-19 surges in the United Kingdom, where distinct variants of SARS-CoV-2 were the dominant stain. We performed a retrospective cohort study investigating critically unwell patients admitted with COVID-19 across three tertiary regional ICUs in London, UK. Of 1782 adult ICU patients screened, 330 intubated and ventilated patients diagnosed with COVID-19 were included. In the second wave where B.1.1.7 variant was the dominant strain, patients were had increased severity of ARDS whilst compliance was greater (p<0.05) and d-dimer lower. The 28-day mortality was not statistically significant (1st wave: 42.2% vs 2nd wave: 39.8%). However, when adjusted for key covariates, the hazard ratio for 28-day mortality in those patients with B.1.1.7 was 3.79 (CI 1.04–13.8; p = 0.043) compared to the original strain. During the second surge in the UK, where the COVID-19 variant B.1.1.7 was most prevalent, significantly more patients presented to critical care with severe ARDS. Furthermore, mortality risk was significantly greater in our ICU population during the second wave of the pandemic in those patients with B.1.1.7. As ICUs are experiencing further waves (particularly by the delta (B.1.617.2) variant), we highlight the urgent need for prospective studies describing immunological and pathophysiological differences across novel emerging variants.

## Introduction

An estimated 5% of patients hospitalized with coronavirus disease 2019 (COVID-19) require critical care unit admission, placing significant burden on global healthcare resources [1–3].

**Data Availability Statement:** All relevant data are within the paper and its Supporting information files.

**Funding:** Our work was funded by the British Journal of Anaesthesia (P89938).

**Competing interests:** The authors have declared that no competing interests exist.

Since the outbreak of COVID-19, caused by severe acute respiratory syndrome coronavirus 2 (SARS-CoV-2), the UK thus far experienced two discrete surges in cases, in keeping with global trends. Distinction between waves in the pandemic is important given that in the UK, the second peak coincided with the identification of a novel variant of concern: B.1.1.7 (termed the Alpha strain). Whilst there is some debate about whether newly emerging variants have a higher mortality than the original SARS-CoV-2 strain, the alpha strain is understood to exhibit a higher natural reproductive number [4, 5].

A number of studies have highlighted ventilator parameters and outcomes from the first surge in critically unwell COVID-19 patients. There is a paucity of data describing emerging variants of SARS-CoV-2, such as the B.1.1.7 (which subsequently progressed to be the dominant strain in many countries), and this is vital to understand any pathological differences between emerging SARS-CoV-2 variants as the pandemic progresses.

The objective of this study was to examine clinical and survival differences in invasively ventilated patients with COVID-19-related ARDS between waves of the pandemic, to help identify important pathophysiological distinctions between variants.

## Materials and methods

This is an observational analysis of all adult patients (≥18 years), performed at three teaching hospitals (within Imperial College NHS trust, London, UK), with the following inclusion criteria in the first 24h after ICU admission: patients sedated and paralysed; requiring invasive mechanical ventilation (with presence of all Berlin definition criteria for ARDS); and confirmed PCR SARS-CoV-2 infection (Fig 1). Physiological or ventilatory variable data was retrospectively collected (at 60minutes following intubation) from patient records. Research ethical approval was not required as this the study was carried out as a service evaluation within the National Health Service (NHS) and recorded under the auspices of the clinical audit office at Imperial College Healthcare NHS Trust and Institutional Data Protection Office.

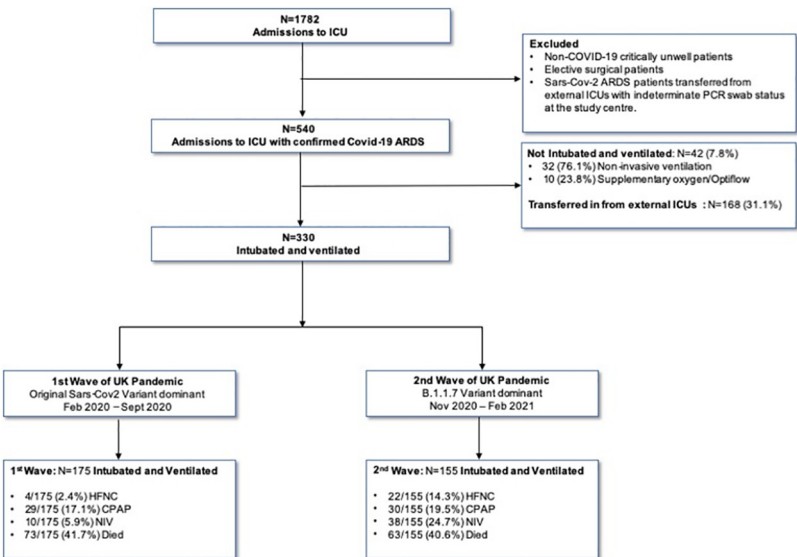

**Fig 1. Study flow diagram demonstrating number of study patients during each wave.** Number of patients receiving high flow nasal oxygen (HFNC), continuous positive airway pressure (CPAP) and non-invasive ventilation (NIV) prior to intubation also recorded.

We defined the first wave as an admission to ICU between 23[rd] February 2020 to 31[st] October 2020, whilst the second wave was 1[st] November 2020 to 23[rd] February 2021, coinciding with the UK Office of National Statistics epidemiological Data [6]. Investigators collecting data were anonymised to those patients who had S-gene target failure (i.e. likely B.1.1.7 infection) [7]. Patients were excluded if they were intubated at hospitals outside of the study (i.e. transferred in from external ICUs). Study patients were followed for 28 days or until they were discharged from the ICU.

Comparison of continuous data between groups was performed using Student's T test or Wilcoxon-Mann-Whitney and comparison of categorical data was done using χ2 or Fisher's exact test. The Kaplan-Meier method and a Cox regression model was used to investigate ICU survival and mortality risk to day 28. The relevant available clinical variable in the adjusted model were sequential organ failure assessment (SOFA) score at ICU admission, sex, and age as previously described [8]. We also accounted for interventions that were used more in the second wave, that may have influenced mortality including application of non-invasive ventilation (NIV) or continuous positive pressure ventilation (CPAP), high flow nasal oxygen, steroids and tocilizumab treatment [9–11]. All analyses were performed using STATA 15 (TX, USA). All statistical tests were two-sided and a p-value $<0.05$ considered significant; due to the exploratory nature of the study, significance level was not adjusted for multiple comparisons.

## Results

Of the 1782 ICU admissions screened, 540 were positive for SARS-CoV-2 and 330 patients met inclusion criteria (1st wave: 175 patients; 2nd wave: 155 patients). Patient demographics are described in Table 1. The severity of ARDS (as per Berlin guidelines) increased in Wave 2. However, dynamic lung compliance was higher (1st wave: 27.5ml/cmH$_2$O vs 2nd wave: 35.1ml/cmH$_2$O, $p<0.05$) and fibrinogen degradation production (d-dimer) was significantly lower (1st wave: 3334ng/ml vs 2nd wave: 2046ng/ml, $p<0.05$) (Fig 2).

We noted in the first wave 95.1% patients accepted for ICU care required intubation and ventilation compared to 82.4% in the second wave ($p<0.05$) (Fig 1). Importantly, the practice of sustained ($>24$hrs continuous use) non-invasive ventilation (NIV) and high-flow nasal cannulae (HFNC) prior to intubation and ventilation for patients increased in the second wave.

ICU Mortality during the second surge was not significantly different (1st wave: 42.2% vs 2nd wave: 39.8%, $p<0.68$) (Table 1) and our data did not demonstrate any survival improvement between pandemic waves (log rank $p = 0.630$) (Fig 3). A Cox proportional hazard model with the first pandemic wave as the reference group, demonstrated a non-significant adjusted hazard ratio (HR) for ICU mortality of 1.39 in the second pandemic wave (95%CI 0.80–2.40, $p = 0.239$; Fig 3, Table 2). However, in patients with variant B.1.1.7, ICU survival was found to be poorer compared to the original variant (log rank $p = 0.004$) with an adjusted HR of 3.79 (CI 1.04–13.8; $p = 0.043$) (Fig 4, Table 3).

## Discussion

This study compares and contrasts clinical parameters and physiological data in patients ventilated with COVID-19 ARDS across two discrete pandemic waves. We show that patients have more severe ARDS in the second surge yet, dynamic lung compliance and fibrinogen degradation production (d-dimer) (a biomarker linked to increased inflammation, vascular endothelial injury and a proposed predictor of poor outcome in COVID-19 [8, 12]) are improved.

When we adjust for the key variables in our cox proportional risk analysis model, we find that in the second wave of the pandemic (despite use of repurposed therapeutics such as

**Table 1. Key demographics, ventilation parameters, treatment and disease severity scores, by UK pandemic wave.**

| | | 1st Wave | 2nd Wave | P value |
|---|---|---|---|---|
| **Demographics** | | N = 175 | N = 155 | |
| **Age—years** | | 59.2 (10.1) | 61.5 (11.2) | 0.045 |
| **Sex—n (%)** | Male | 125/175 (72.1%) | 87/155 (56.5%) | 0.005 |
| **BMI—kg/m²** | | 28.4 (5.5) | 29.4 (6.6) | 0.133 |
| **Ethnicity** | White | 28/175 (16.0%) | 37/155 (23.9%) | 0.073 |
| **Disease** | | | | |
| **ARDS Severity at intubation—n (%)** | Mild | 29/175 (16.6%) | 15/155 (9.7%) | 0.013 |
| | Moderate | 71/175 (40.6%) | 63/155 (40.9%) | .. |
| | Severe | 75/175 (42.9%) | 75/155 (48.7%) | .. |
| **PF ratio—kPa** | | 18.1 (10.6) | 14.1 (6.3) | 0.004 |
| **$C_{rs}$—ml/cmH$_2$O** | | 27.3 (14.0) | 33.8 (22.8) | 0.007 |
| **D-dimer in first 24hr—ng/ml (IQR)** | | 3334 (1545–9215) | 2046 (1076–4580) | 0.006 |
| **ICU length of stay—days (IQR)** | | 18.7 (16.4) | 12.1 (14.0) | 0.004 |
| **HFNC$^\$$—n (%)** | | 4/175 (2.4%) | 22/155 (14.3%) | <0.001 |
| **CPAP$^\$$—n (%)** | | 29/175 (17.1%) | 30/155 (19.5%) | 0.573 |
| **NIV$^\$$—n (%)** | | 10/175 (5.9%) | 38/155 (24.7%) | <0.001 |
| **ICU Mortality—n (%)** | | 73/175 (42.2%) | 63/155 (40.9%) | 0.810 |
| **Disease severity scores** | | | | |
| **APACHE II** | | 14.9 (5.4) | 17.0 (6.9) | 0.003 |
| **SOFA Score** | | 6.4 (2.9) | 6.1 (3.4) | 0.313 |
| **Treatments** | | | | |
| **Steroids—n (%)** | | 89/175 (50.9%) | 153/155 (99.2%) | <0.001 |
| **Tocaluzimab—n (%)** | | 11/175 (6.3%) | 49/155 (31.8%) | <0.001 |
| **Remdesivir—n (%)** | | 5/175(2.9%) | 64/155 (41.6%) | <0.001 |

Abbreviations: APACHE II, Acute Physiology And Chronic Health Evaluation II; BMI, Body Mass Index; CPAP, Continuous Positive Airways Pressure; $C_{rs}$, static compliance; HFNC, High Flow Nasal Cannulae; ICU, intensive care unit; IQR, interquartile range; n, number; NIV, Non-invasive ventilation; PF ratio, PaO2/FiO2 ratio; SOFA, Sequential organ failure assessment.

$^\$$ denotes treatment administered for >24hrs continuously, prior to mechanical ventilation.

Patient demographics and clinical data are reported as mean (standard deviation), median (inter-quartile range) or as percentages. Comparison of continuous data between groups was done using Student's T test or Wilcoxon-Mann-Whitney and comparison of categorical data was done using χ2 or Fisher's exact test as appropriate.

corticosteroids and Tocilizumab) [13–15], there is no improvement in ICU mortality. This was particularly surprising considering the benefit of tocilizumab and corticosteroids in reducing COVID-19 mortality is well established [10, 11]. The ongoing emergence of novel variants of interest is one plausible explanation for the increased risk. We found poorer survival in individuals with B.1.1.7 with a 3.79-fold increase in mortality risk compared to non-B.1.1.7 (albeit with a large confidence interval). This finding contrasts with a previous report where no difference in ICU mortality was found between B.1.1.7 and non-B.1.1.7 COVID-19 patients. However, this study did not adjust for key covariates discussed above and may explain this discrepancy. Indeed, data from the Intensive Care National Audit and Research Centre does suggest a trend towards higher mortality during the second surge in the UK but did not specifically examine the mortality between different variants [16]. Indeed emerging data does also suggest and increased mortality risk from B.1.1.7 consistent with the data presented in this report [17]. It may also be prudent to evaluate whether treatments found to have significant mortality benefit with older COVID-19 variants, continue to be successful as variants mutate and pathophysiology potentially alters.

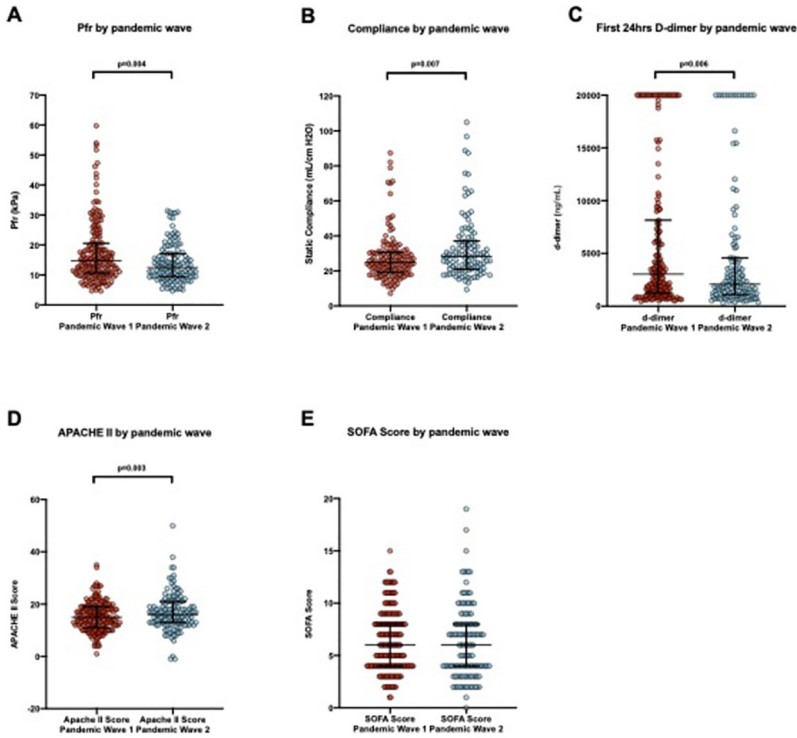

**Fig 2. Ventilator, biochemical and physiological data.** A-E) demonstrates the change in static compliance, Pfr, plasma d-dimer, APACHE II and SOFA score respectively. Upper limit for d-dimer concentration is 20,000ng/mL.

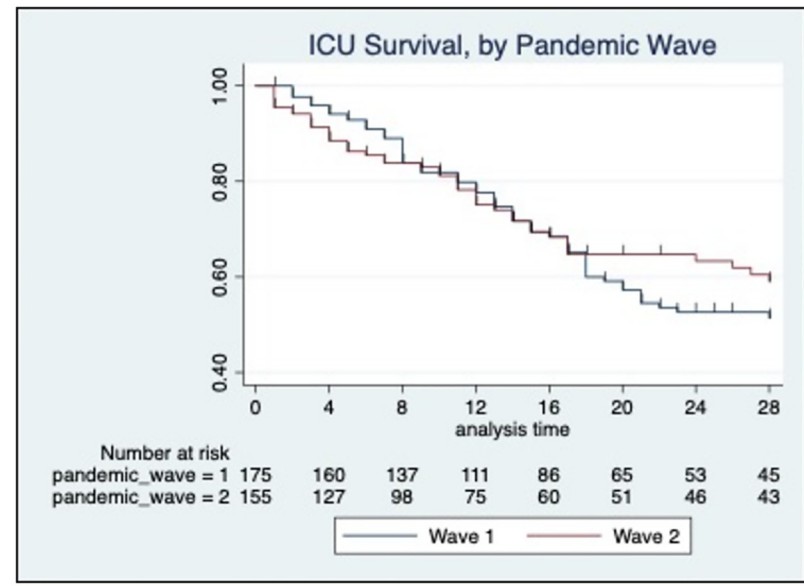

**Fig 3. Kaplan-Meier survival demonstrates the proportion of ICU survival by UK pandemic wave.** Single hash mark denotes an ICU discharge (alive). Log-rank p = 0.630 demonstrating no statistical difference in survival between the pandemic waves.

**Table 2. Cox proportional risk analysis for mortality of all intubated and ventilated patients across both UK waves of the Sars-Cov-2 pandemic.**

| Covariates | | Hazard ratio (95% CI) | P value |
|---|---|---|---|
| Pandemic Wave | | | |
| | First Wave | 1 (ref) | |
| | Second Wave | 1.74 (1.05–2.89) | 0.035 |
| Sex | | | |
| | Male | 1 (ref) | |
| | Female | 0.67 (0.43–1.06) | 0.241 |
| Age | | 1.04 (1.02–1.06) | < 0.001 |
| Corticosteroid treatment | | 0.46 (0.27–0.78) | 0.016 |
| Tocilizumab treatment | | 1.36 (0.77–2.38) | 0.875 |
| High Flow Nasal Oxygen | | 1.76 (0.92–3.36) | 0.779 |
| NIV/CPAP | | 0.66 (0.42–1.04) | 0.210 |
| SOFA Score | | 1.04 (0.98–1.11) | 0.031 |

Abbreviations: CI, confidence interval; SOFA, Sequential organ failure assessment.

Given the retrospective nature of this study and the relatively short experience with this novel disease, there are likely to be a host of unknown confounding factors affecting the interpretation of these results. A possible contribution is that by the second wave, experience managing these critically unwell patients had become greater, which dictated changes to practice, for example, increased administration of NIV and HFNC on acute respiratory units/medical wards which in turn are likely to have altered the population referred to ICU. As well as changes to patient selection, frailty and comordities, the timing of mechanical ventilation may also have differed in the disease process. We have included d-dimer in our analysis as it has been shown to correlate with disease severity and is an accurate biomarker for predicting

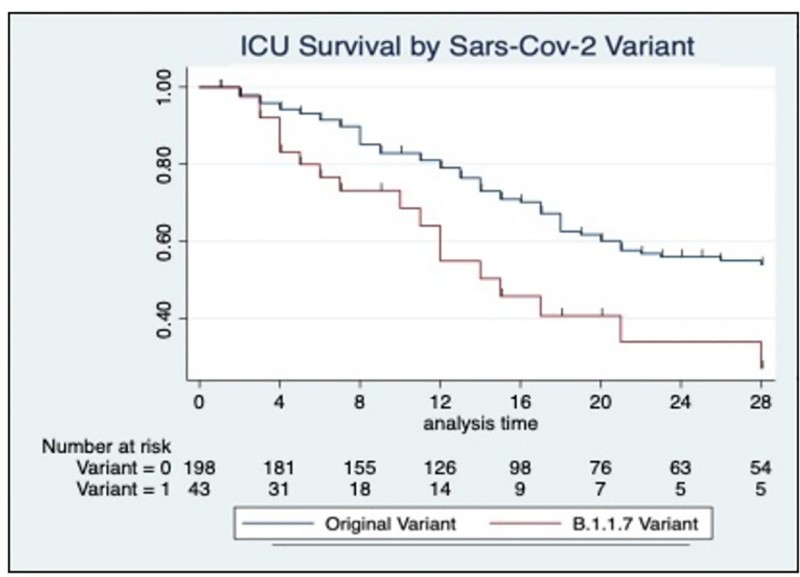

**Fig 4. Kaplan-Meier survival demonstrates the proportion of ICU survival by Variant.** Log-rank p = 0.004 demonstrating that those patients with variant B.1.1.7 have a significant less chance of survival compared to non-B.1.1.7 variants.

**Table 3. Cox proportional risk analysis for mortality of all intubated and ventilated patients by B.1.1.7 and the original variant.**

| Covariates | | Hazard ratio (95% CI) |
|---|---|---|
| Variant | | |
| | Original variant | 1 (ref) |
| | B.1.1.7 variant | 3.79 (1.04–13.83) |
| Pandemic Wave | | |
| | First Wave | 1 (ref) |
| | Second Wave | 0.39 (0.11–1.31) |
| Sex | | |
| | Male | 1 (ref) |
| | Female | 0.93 (0.54–1.60) |
| Age | | 1.04 (1.02–1.07) |
| Corticosteroid treatment | | 0.55 (0.32–0.95) |
| APACHE II Score | | 1.06 (1.00–1.12) |
| SOFA Score | | 0.94 (0.85–1.05) |

Abbreviations: APACHE II, Acute Physiology And Chronic Health Evaluation II; CI, confidence interval; SOFA, Sequential organ failure assessment. The adjusted for patient-level factors; age, sex, Sequential (Sepsis-Related) Organ Failure Assessment (SOFA) score, APACHE II, Corticosteroid treatment—selected based on our study hypothesis, existing literature [8] and hypothesized associations with mortality.

mortality in COVID-19 [8, 12]. However we found that in those patients with B.1.1.7, d-dimer levels are improved compared to non-B.1.1.7 patients, despite the increase in mortality. There are a number of potentially reasons for this including the effect of treatment such as steroids or tocilizumab. Our data suggests that a re-evaluation is required of various biomarkers that were initially thought to be predictive of disease severity as the pandemic progresses, particularly with as newer variants are discovered and novel treatments are introduced into clinical practice. For instance neutrophil/lymphocyte ratio was been associated with disease severity with the original SARS-COV-2 variant due to significant reduction in lymphocytes and increase of neutrophils in patients with severe COVID during this phase of the pandemic. However this has not been replicated with subsequent waves and counts of neutrophils and lymphocytes may not change significantly between different severity groups [18].

There are some limitations to our report. Firstly, the ventilator settings were not protocolised instead judged to be most appropriate by the senior clinician following intubation. In addition, some patients may have deteriorated post hospital discharge (e.g. at home) and whilst this is likely to be similar in both waves, their outcome is not captured. Thirdly, whilst we considered the covariates to include in our analysis, we did not account for other co-morbidities such as cardiovascular disease [19], cancer [20] and chronic kidney disease [21], all of which have been associated with worse outcomes in COVID-19. Fourthly, scoring systems based upon computer tomography (CT) have been used to characterize COVID-19 severity and outcomes (e.g. requirement of mechanical ventilation/mortality) in patients, particularly on presentation to hospital [22]. CT scores were not included in this study as all patients were already intubated and ventilated. Furthermore, we are unable to comment if ventilatory data at time of intubation correlated with CT score as most patients had CT scans when clinically stable to undertake transfer to radiology departments, rather than specifically at the onset of mechanical ventilation. Finally, cases of SARS-CoV-2 with an S gene target failure were identified as highly suspicious of B.1.1.7 and we did not perform complete genomic sequencing on all patient samples. However S gene target failure has been shown to be a reliably distinguish

variant B.1.1.7 from the original SARS-CoV-2 strain and has been used as such in a number of studies [23, 24].

Nevertheless, this study does yield important insights into the changing characteristics and mortality within the ICU population across the first two waves of the pandemic. Although mortality risk in patients with COVID-19 is tempered by corticosteroids and tocilizumab, our data highlight that the alpha variant may result in a higher risk of death compared to the original variant. As critical care units experience further waves (particularly those affected by the delta (B.1.617.2) variant), we highlight the urgent need for further controlled prospective studies describing the immunological and pathophysiological differences across novel emerging variants, which will improve mechanistic understanding of COVID-19 and potentially identify novel therapeutic targets.

## Supporting information

**S1 Data.**
(XLSX)

## Acknowledgments

We would like to thank all the clinical staff and the Clinical Audit Teams at Hammersmith Hospital, St Mary's Hospital and Charing Cross Hospital and the Clinical Audit Team for their assistance with collecting the clinical data.

## Author Contributions

**Conceptualization:** Owais Kadwani, Elizabeth Brown, Richard Stümpfle, Parind Patel, Stephen J. Brett, Sanooj Soni.

**Data curation:** Andrew I. Ritchie, Owais Kadwani, Dina Saleh, Lesley R. Broomhead, Paul Randell, Maie Templeton, Stephen J. Brett, Sanooj Soni.

**Formal analysis:** Andrew I. Ritchie, Owais Kadwani, Dina Saleh, Lesley R. Broomhead, Paul Randell, Sanooj Soni.

**Investigation:** Behrad Baharlo, Sanooj Soni.

**Methodology:** Andrew I. Ritchie, Behrad Baharlo, Umeer Waheed, Stephen J. Brett, Sanooj Soni.

**Project administration:** Sanooj Soni.

**Software:** Andrew I. Ritchie.

**Supervision:** Behrad Baharlo.

**Writing – original draft:** Andrew I. Ritchie, Owais Kadwani, Stephen J. Brett, Sanooj Soni.

**Writing – review & editing:** Andrew I. Ritchie, Stephen J. Brett, Sanooj Soni.

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
