## [Decision Letter · Decision Letter 0]

20 Dec 2021

PONE-D-21-31797Clinical and Survival differences during separate COVID-19 surges: Investigating the impact of the Sars-CoV-2 alpha variant in critical care patientsPLOS ONE

Dear Dr. Soni,

Thank you for submitting your manuscript to PLOS ONE. After careful consideration, we feel that it has merit but does not fully meet PLOS ONE’s publication criteria as it currently stands. Therefore, we invite you to submit a revised version of the manuscript that addresses the points raised during the review process. Editor's comments:

1. Please follow the PLoS One's guidelines for manuscript preparation.

2. Please cite at least 5 articles from PLoS One.

3. Please revise the manuscript per the following reviewers' comments.

We look forward to receiving your revised manuscript.

Kind regards,

Farzad Taghizadeh-Hesary

Academic Editor

PLOS ONE

Journal Requirements:

Reviewers' comments:

Reviewer's Responses to Questions

**Comments to the Author**

1. Is the manuscript technically sound, and do the data support the conclusions?

Reviewer #1: Yes

Reviewer #2: No

Reviewer #3: Partly

2. Has the statistical analysis been performed appropriately and rigorously? 

Reviewer #1: Yes

Reviewer #2: No

Reviewer #3: Yes

3. Have the authors made all data underlying the findings in their manuscript fully available?

Reviewer #1: Yes

Reviewer #2: No

Reviewer #3: Yes

4. Is the manuscript presented in an intelligible fashion and written in standard English?

Reviewer #1: Yes

Reviewer #2: Yes

Reviewer #3: No

5. Review Comments to the Author

Reviewer #1: This is a retrospective cohort study, as such, the results are primarily hypothesis-generating. Nevertheless, it is interesting that there is no difference in mortality between the two waves even though the patients in the second wave have largely received treatment that has been documented to have an effect on survival. This may indicate that the characteristics of the virus could be of importance. The article describes examples of what are called key biomarkers. Here it is appropriate to remind that our designation of what are important markers, is largely influenced by what we can measure more than by the fact that it has a decisive role in the process we seek to describe.

The actual division into two distinct virus variants is not based on a complete gene sequencing, but a so-called S gene target failure. How precise this division is, is not explained. Can the authors comment on this?

The authors address a number of important limitations of this type of study, not least it is pointed out that it is to be expected that there are likely to be a host of unknown confounding factors affecting the interpretation of these results.

However, this work contains observations that will be very interesting to clarify further, and as the authors say, it will be necessary to carry out controlled prospective studies in order to be able to determine the significance of different virus variants on the course of the disease and death.

Reviewer #2: Although the study addressed an interesting topic in the COVID-19 periods, the study design preclude the conclusion to be drawn with current analysis.

1. while the authors aimed to compare difference in two variants of the virus, the difference in current analysis can be due to other confounding factors that cannot be fully excluded.

2. The 2d wave patients showed more severe illness than at baseline than the 1st wave, which cannot be fully due to the virus variants. Other factors such as availability of ICU beds, older age and more comorbidity burden can cause this phenomenon.

3. It is interesting to note that nearly all patients in the 2d wave were treated with steroids while the usefulness of steoids were not confirmed. you need discuss this issue since there has been many literature on this issue (Crit Care. 2020 Dec 18;24(1):698. doi: 10.1186/s13054-020-03429-w. Ann Intensive Care. 2021 Nov 26;11(1):159. doi: 10.1186/s13613-021-00951-0.).

4. "1st wave: 42.2% vs 2nd wave: 39.8%, p<0.68"--such reporting of p<0.68 is not meaningful in the text.

5. I also suggest to report p value in table 2.

6. The confidence interval for B.1.1.7 variant in table 3 is very large, should interpret with caution.

7. There is no description of how variables were selected to enter into the multiple regression model, you need to clarify. there are many methods such as purposeful selection, stepwise and so on. But the authors fail to prespecify the model.

Reviewer #3: Author present their work as "Clinical and Survival differences during separate COVID-19 surges: Investigating the

impact of the Sars-CoV-2 alpha variant in critical care patients"

The manuscript is more of audit analysis of retrospective data.

Many similar articles now have been already published in recent times

1. Tracking the Emergence of SARS-CoV-2 Alpha Variant in the United Kingdom NEJM

2. The Disease Severity and Clinical Outcomes of the SARS-CoV-2 Variants of Concern by Lin et al.- Frontiers in Public Health

3. Mortality and critical care unit admission associated with the SARS-CoV-2 lineage B.1.1.7 in England: an observational cohort study. Lancet Infect Dis. 2021 by Patone et al.

The intent is good however the methodology is unclear. Inclusion criteria says “requiring invasive mechanical ventilation” whereas Fig 1 Flowchart includes NIV patients along with HFNC patients. On one hand authors say “Physiological or ventilatory variable data” and on other hand ventilator settings and protocol were probably different which can be responsible for variations in observations.

It is agreed about predominant variant; however was the alpha variant confirmed ? It is listed as limitation but should have confirmed for research article. Most data is on presumption of SARS-CoV-2 lineage B.1.1.7.

“with B.1.1.7. As ICUs are experiencing further waves (particularly by the delta (B.1.617.2) variant”

It appears Delta variant is on the wane, Which variants are now of concern or active in UK ?

There are many inflammatory variables studied. Why particular two variables selected- what about TLC, N/L ratio, CRP, IL-6 ? These markers also have shown co-relation in other studies.

What was the criteria of prescribing Toculizumab ?

The ventilator settings may also be dependent on amount of lung parenchyma involvement i.e CT Score. Any correlation with CT score ?

The discussion is not up to the mark. It needs to be worked upon with inputs from similar articles.

6. PLOS authors have the option to publish the peer review history of their article (what does this mean?). If published, this will include your full peer review and any attached files.

Reviewer #1: No

Reviewer #2: No

Reviewer #3: No

---

## [Author Response · Author response to Decision Letter 0]

15 Feb 2022

Responses to Reviewers

We would like to thank the editor for the opportunity to respond to the reviewers’ comments and are very grateful to the reviewers for their time in evaluating our paper. We have provided responses below to all of the concerns raised.

Editor's comments:

Comment 1: ‘Please follow the PLoS One's guidelines for manuscript preparation.’

Response 1: We have now amended the manuscript in line with PLoS One’s guidelines 

Comment 2: ‘Please cite at least 5 articles from PLoS One’.

Response 2: We have added 5 Citations relating to PlosOne articles

Comment 3: ‘Please revise the manuscript per the following reviewers' comments’.

Response 3: Thank you, we have carefully considered all the comments made by the reviewers which we thank for being constructive. Please see our point by point responses below.

Reviewer #1

We thank the reviewer for his/her time and detailed, helpful comments and below are point-by-point responses to each of these. We have implemented the majority of the suggestions made by this reviewer, and provided explanations where we do not agree.

Comment 1: ‘This is a retrospective cohort study, as such, the results are primarily hypothesis-generating. Nevertheless, it is interesting that there is no difference in mortality between the two waves even though the patients in the second wave have largely received treatment that has been documented to have an effect on survival. This may indicate that the characteristics of the virus could be of importance. The article describes examples of what are called key biomarkers. Here it is appropriate to remind that our designation of what are important markers, is largely influenced by what we can measure more than by the fact that it has a decisive role in the process we seek to describe.’

Response 1: We appreciate these comments and review of the article.

Comment 2: ‘The actual division into two distinct virus variants is not based on a complete gene sequencing, but a so-called S gene target failure. How precise this division is, is not explained. Can the authors comment on this?’

Response 2: We thank the reviewer for this point. It has been shown comprehensively that S gene target failure (SGTF) was a reliable marker of B.1.1.7 (Brown et al JAMA 2021; doi 10.1001/jama.2021.5607). Indeed SGTF had been consistently used to identify B.1.1.7 variant in a number of high profile studies (e.g. Walker et al NEJM 2021; doi 10.1056/NEJMc2103227) and therefore we feel that this is a precise marker to differentiate alpha variant from the original SARS-CoV-2 variant.

We have added the following phrase to the discussion/limitations:

‘However S gene target failure has been shown to be a reliably distinguish variant B.1.1.7 from the original SARS-CoV-2 strain and has been used as such in a number of studies.’ 

Comment 3: ‘The authors address a number of important limitations of this type of study, not least it is pointed out that it is to be expected that there are likely to be a host of unknown confounding factors affecting the interpretation of these results. However, this work contains observations that will be very interesting to clarify further, and as the authors say, it will be necessary to carry out controlled prospective studies in order to be able to determine the significance of different virus variants on the course of the disease and death.’

Response 3: Thank you for your comments regarding our discussion and conclusion. We agree and have added “controlled” to the relevant sentence in the discussion to better emphasise this point.

“we highlight the urgent need for further controlled prospective studies describing the immunological and pathophysiological differences across novel emerging variants, which will improve mechanistic understanding of COVID-19 and potentially identify novel therapeutic targets.”

Reviewer #2

We thank the reviewer for his/her time and detailed, helpful comments and below are point-by-point responses to each of these. We have implemented the majority of the suggestions made by this reviewer, and provided explanations where we do not agree.

Comment 1: ‘while the authors aimed to compare difference in two variants of the virus, the difference in current analysis can be due to other confounding factors that cannot be fully excluded.’

Response 1: We agree with that the nature of the study design does not preclude unknown confounding factors. Variables included in our cox regression were decided a priori to allow adjustment for known confounding factors based on the existing literature (PMID: 32861276). We acknowledge this with the sentence: 

“there are likely to be a host of unknown confounding factors affecting the interpretation of these results”

Comment 2: ‘The 2nd wave patients showed more severe illness than at baseline than the 1st wave, which cannot be fully due to the virus variants. Other factors such as availability of ICU beds, older age and more comorbidity burden can cause this phenomenon.’

Response 2: Thank you for this comment and we agree with this interesting observation. We included covariates such as SOFA score and use of NIV/CPAP or HRNC oxygen support prior to intubation, but have amended our discussion to acknowledge further factors which may have accounted for our observation.

“As well as changes to patient selection, frailty and co-morbidities, the timing of mechanical ventilation may also have differed in the disease process.”

Comment 3: ‘It is interesting to note that nearly all patients in the 2d wave were treated with steroids while the usefulness of steroids were not confirmed. you need discuss this issue since there has been many literature on this issue (Crit Care. 2020 Dec 18;24(1):698. doi: 10.1186/s13054-020-03429-w. Ann Intensive Care. 2021 Nov 26;11(1):159. doi: 10.1186/s13613-021-00951-0.).’

Response 3: Thank you for this comment and including two interesting studies on the subject of corticosteroid use in the ICU setting. The second wave of the pandemic in the UK coincided with a change to UK Covid-19 treatment guidelines which stipulated that dexamethasone should be offered to all hospitalised patients requiring oxygen supplementation. Patients in the first wave were treated off license or as part of open label clinical trials such as “RECOVERY”. However, we very respectfully disagree with the reviewer. Our data does not question the usefulness of corticosteroids, rather demonstrate that despite their use, mortality in the second wave remains very high. Our data suggest that this is because of an increased risk of mortality with B.1.1.7 and we have discussed in the discussion:

‘The ongoing emergence of novel variants of interest is one plausible explanation for the increased risk. We found poorer survival in individuals with B.1.1.7 with a 3.79-fold increase in mortality risk compared to non-B.1.1.7’

However, in light of the reviewer’s comments, we have added in the final paragraph:

“we highlight the urgent need for further controlled prospective studies describing the immunological and pathophysiological differences across novel emerging variants, which will improve mechanistic understanding of COVID-19 and potentially identify novel therapeutic targets.”

Comment 4: ‘1st wave: 42.2% vs 2nd wave: 39.8%, p<0.68"--such reporting of p<0.68 is not meaningful in the text’.

Response 4: Thank you we have removed this. 

Comment 5: I also suggest to report p value in table 2.

Response 5: Thank you we have made this addition to table 2. 

Comment 6: ‘The confidence interval for B.1.1.7 variant in table 3 is very large, should interpret with caution’ 

Response 6: Thank you for this comment and we will draw attention to this in our discussion.

“We found poorer survival in individuals with B.1.1.7 with a 3.79-fold increase in mortality risk compared to non-B.1.1.7 (albeit with a large confidence interval).”

Comment 7: ‘There is no description of how variables were selected to enter into the multiple regression model, you need to clarify. there are many methods such as purposeful selection, stepwise and so on. But the authors fail to prespecify the model.’

Response 7: As mentioned in the Response to comment 1, variables included in our cox regression were decided a priori to allow adjustment for known confounding factors based on the existing literature (PMID: 32861276). In acknowledgement of the reviewer’s comments, we have added the following section into the methods:

‘The relevant available clinical variable in the adjusted model were sequential organ failure assessment (SOFA) score at ICU admission, sex, and age as previously described. We also accounted for interventions that were used more in the second wave, that may have influenced mortality including application of non invasive ventilation (NIV) or continuous positive pressure ventilation (CPAP), high flow nasal oxygen steroids and tocilizumab treatment’

Reviewer #3: 

We thank the reviewer for his/her time and detailed, helpful comments and below are point-by-point responses to each of these. We have implemented the majority of the suggestions made by this reviewer, and provided explanations where we do not agree.

Comment 1: ‘Author present their work as "Clinical and Survival differences during separate COVID-19 surges: Investigating the impact of the Sars-CoV-2 alpha variant in critical care patients"

The manuscript is more of audit analysis of retrospective data.

Many similar articles now have been already published in recent times

1. Tracking the Emergence of SARS-CoV-2 Alpha Variant in the United Kingdom NEJM

2. The Disease Severity and Clinical Outcomes of the SARS-CoV-2 Variants of Concern by Lin et al.- Frontiers in Public Health

3. Mortality and critical care unit admission associated with the SARS-CoV-2 lineage B.1.1.7 in England: an observational cohort study. Lancet Infect Dis. 2021 by Patone et al.’

Response 1: We thank the reviewer for their comment and have stated in the methods that the report was a carried out as a service evaluation. We have reference these papers in the discussion.

Comment 2: ‘The intent is good however the methodology is unclear. Inclusion criteria says “requiring invasive mechanical ventilation” whereas Fig 1 Flowchart includes NIV patients along with HFNC patients. On one hand authors say “Physiological or ventilatory variable data” and on other hand ventilator settings and protocol were probably different which can be responsible for variations in observations.’

Response 2: We sincerely apologise for the confusion and have made the legend for Figure 1 clearer. The flow chart mentions NIV and HFNC but this is to explain the percentage of patients who received these therapies prior to intubation. By the second wave, there was an increased administration of NIV and HFNC on acute respiratory units/medical wards which likely altered the population referred to ICU. Our intention is to recognise and account for this as could well have affected ICU mortality. 

‘Fig 1: Study flow diagram demonstrating number of study patients during each wave. Number of patients receiving high flow nasal oxygen (HFNC), continuous positive airway pressure (CPAP) and non-invasive ventilation (NIV) prior to intubation also recorded.’ 

Comment 3: ‘It is agreed about predominant variant; however was the alpha variant confirmed ? It is listed as limitation but should have confirmed for research article. Most data is on presumption of SARS-CoV-2 lineage B.1.1.7.’

Response 3: We thank the reviewer for this point. It has been shown comprehensively that S gene target failure (SGTF) was a reliable marker of B.1.1.7 (Brown et al JAMA 2021; doi 10.1001/jama.2021.5607). Indeed SGTF had been consistently used as a measure of B.1.1.7 variant in a number of high profile studies (e.g. Walker et al NEJM 2021; doi 10.1056/NEJMc2103227) and therefore we feel that this is a precise marker to differentiate alpha variant from the original SARS-CoV-2 variant.

Comment 4: ‘with B.1.1.7. As ICUs are experiencing further waves (particularly by the delta (B.1.617.2) variant”. It appears Delta variant is on the wane, Which variants are now of concern or active in UK?’

Response 4: Omicron is the spreading rapidly within the UK and is the active variant.

Comment 5: ‘There are many inflammatory variables studied. Why particular two variables selected- what about TLC, N/L ratio, CRP, IL-6 ? These markers also have shown co-relation in other studies.’

Response 5: We thank the reviewer for these helpful comments. We chose PF ratio and dynamic compliance as these are easy to measure bedside tests that accurately describe the severity of ARDS and extent of disease burden on the lungs. 

D-dimer was chosen as a measure of hypercoagulability and a systematic review (Varikusuvu et al 2021) of 113 studies demonstrated that D-dimer was accurate for predicting disease progression in COVID-19. Indeed an article published in this journal (Poudel et al) demonstrated that d-dimer on admission is an accurate biomarker for predicting mortality in COVID. 

We did not include the variables suggested by the reviewer (total lymphocyte, neutrophil/lymphocyte (N/L ratio), CRP and IL-6) as we were not convinced of the diagnostic value of these variables. For example, it has been found that NLR correlated with disease severity during the 1st wave in COVID-19 patients due to the significant reduction of lymphocyte and the significant increase of neutrophil in severe COVID-19 patients. However, already published data confirms that this was not replicated in the 2nd wave and the number of lymphocytes and neutrophils did not change significantly between patients of different severity subgroups. Gelzo et al (2021 Frontiers in Immunology) hypothesise that this difference was likely because of the steroid therapy between the first and second wave. Furthermore, Gelzo as shows that biomarkers such as serum interleukin IL-6, which gradually increased with disease severity in patients of the 1st wave, did not correlate with disease severity in the 2nd wave when alpha variant was dominant. There has been a significant amount of debate about Il-6 levels and a review demonstrated that its levels are much lower than hypo or hyperinflammatory phenotypes of ARDS (Sinha et al 2020). 

Finally CRP is known to dramatically supressed in patients who have been given tocilizumab (Hofmaenner et al 2021) and have very little and have very little diagnostic yield. Therefore we are unsure of the relevance of reporting it in our study as it may provide misleading results.

However in acknowledgement of the reviewers comments, we have made the following changes in the discussion:

- We have removed the term ‘when we adjust for key biomarker’ and changed to ‘when we adjust for the key variable in our cox proportional; risk analysis model’ (line 186)

- ‘We have included d-dimer in our analysis as it has been shown to correlate with disease severity and is an accurate biomarker for predicting mortality in COVID-19. However we found that in those patients with B.1.1.7, d-dimer levels are improved compared to non-B.1.1.7 patients, despite the increase in mortality. There are a number of potentially reasons for this including the effect of treatment such as steroids or tocilizumab. Our data suggests that a re-evaluation is required of various biomarkers that were initially thought to be predictive of disease severity as the pandemic progresses, particularly with as newer variants are discovered and novel treatments are introduced into clinical practice. For instance neutrophil/lymphocyte ratio was been associated with disease severity with the original SARS-COV-2 variant due to significant reduction in lymphocytes and increase of neutrophils in patients with severe COVID during this phase of the pandemic. However this has not been replicated with subsequent waves and counts of neutrophils and lymphocytes may not change significantly between different severity groups.’ 

Comment 6: ‘What was the criteria of prescribing Toculizumab?’

Response 6: The REMAP CAP trial reported their results of tocilizumab in Jan 2021 and prior to then, patients were prescribed it as part of the trial. Thereafter, each patient was discussed on a case by case bass by the Imperial College Healthcare NHS Trust independent COVID-19 Treatment Group (made up of specialists in the field). 

Comment 7: The ventilator settings may also be dependent on amount of lung parenchyma involvement i.e CT Score. Any correlation with CT score ?

Response 7: This is a very interesting comment and we thank the reviewer for this. However sadly we were unable to do this. For our patients, there was no protocol for when precisely a patient should have CT imaging of their lungs (meaning that patients had scans performed at different times during their disease course). For instance most patients had CT scans when clinically stable to undertake the transfer to radiology department. When patients were intubated, they were often very unstable and not able to tolerate this transfer (and therefore no accompanying CT scan for that day). As a ventilatory data was collected at time of intubation, we are unable to comment if the ventilatory data at time of intubation correlated with CT score. In addition, not all our patients had CT scans, particularly if they were too unwell to transfer for a CT scan and subsequently died (particularly during the first wave).

Comment 8:. The discussion is not up to the mark. It needs to be worked upon with inputs from similar articles.

Response 8: We thank the reviewer for pointing this out. We have now made a number of changes to the discussion, which we hope brings it up to expected standard. We have also referenced all the aforementioned papers discussed by the reviewer, which have similar inputs.

---

## [Decision Letter · Decision Letter 1]

21 Apr 2022

PONE-D-21-31797R1Clinical and Survival differences during separate COVID-19 surges: Investigating the impact of the Sars-CoV-2 alpha variant in critical care patientsPLOS ONE

Dear Dr. Soni,

Thank you for submitting your manuscript to PLOS ONE. After careful consideration, we feel that it has merit but does not fully meet PLOS ONE’s publication criteria as it currently stands. Therefore, we invite you to submit a revised version of the manuscript that addresses the points raised during the review process.

We look forward to receiving your revised manuscript.

Kind regards,

Farzad Taghizadeh-Hesary

Academic Editor

PLOS ONE

Journal Requirements:

Additional Editor Comments:

Editor:

1. Please find the reviewer's 2 comments for the remaining revision.

2. Please prepare the tables per the standard "AMA Manual of Style".

3. There is strong evidence denoting the importance of comorbidities on the survival of patients with COVID-19, such as cardiovascular disease (https://academic.oup.com/eurjpc/article/28/14/1599/6145858), cancer (https://brieflands.com/articles/ijcm-110907.html, https://www.ncbi.nlm.nih.gov/pmc/articles/PMC8184167/), chronic kidney disease (https://journals.plos.org/plosone/article?id=10.1371/journal.pone.0254525#:~:text=The%20crude%20mortality%20rate%20among,19%20in%20the%20PIRP%20cohort.), among others. I found that the authors have not evaluated this factor. I suggest considering this factor in the analysis; if the data are not available, please address it in the limitations by using the mentioned references.

Reviewer 1: Well done; my previous comments are well addressed and I am satisfied with these amendments. my previous comments are well addressed and I am satisfied with these amendments.

Reviewer 2: 1.Though the authors have improved the manuscript draft; There are still few confounding variables which could have impacted the outcome in second wave that needs to be evaluated and mentioned in the limitations. (e.g., CT score, day of presentation)

2. The authors mention "Although mortality risk in patients with COVID-19 is tempered by corticosteroids and tocilizumab" however fail to address this in the discussion.

Steroids – (50.9%) 1st wave vs (99.2%) of second wave

Tocilizumab –(6.3%) in 1st wave and (31.8%) of second wave

Remdesivir – (2.9%) in 1st wave and (41.6%) of second wave.

3. The discussion still needs improvement based on the above facts

Reviewer 3:

I am satisfied with the answers provided by the authors, and I have no further comments. This is acceptable.

Reviewers' comments:

Reviewer's Responses to Questions

**Comments to the Author**

1. If the authors have adequately addressed your comments raised in a previous round of review and you feel that this manuscript is now acceptable for publication, you may indicate that here to bypass the “Comments to the Author” section, enter your conflict of interest statement in the “Confidential to Editor” section, and submit your "Accept" recommendation.

Reviewer #1: All comments have been addressed

Reviewer #2: All comments have been addressed

Reviewer #3: (No Response)

2. Is the manuscript technically sound, and do the data support the conclusions?

Reviewer #1: Yes

Reviewer #2: Yes

Reviewer #3: Partly

3. Has the statistical analysis been performed appropriately and rigorously? 

Reviewer #1: Yes

Reviewer #2: Yes

Reviewer #3: Yes

4. Have the authors made all data underlying the findings in their manuscript fully available?

Reviewer #1: Yes

Reviewer #2: Yes

Reviewer #3: Yes

5. Is the manuscript presented in an intelligible fashion and written in standard English?

Reviewer #1: Yes

Reviewer #2: Yes

Reviewer #3: Yes

6. Review Comments to the Author

Reviewer #1: I am satisfied with the answers provided by the authors, and I have no further comments. This is acceptable

Reviewer #2: well done; my previous comments are well addressed and I am satisfied with these amendaments

my previous comments are well addressed and I am satisfied with these amendaments

Reviewer #3: (No Response)

7. PLOS authors have the option to publish the peer review history of their article (what does this mean?). If published, this will include your full peer review and any attached files.

Reviewer #1: No

Reviewer #2: **Yes: **Zhongheng Zhang

Reviewer #3: No

---

## [Author Response · Author response to Decision Letter 1]

4 May 2022

Responses to Reviewers

We would like to thank the editor for the opportunity to respond to the reviewers’ comments and are very grateful to the reviewers for their time in evaluating our paper. We have provided responses below to all of the concerns raised.

Editor's comments:

Comment 1: ‘Please find the reviewer's 2 comments for the remaining revision.’

Response 1: We have responded to all of reviewer’s 2 comennts for the remaining revision 

Comment 2: ‘Please prepare the tables per the standard "AMA Manual of Style".

Response 2: We have prepared the tables as per the requested standard

Comment 3: ‘There is strong evidence denoting the importance of comorbidities on the survival of patients with COVID-19, such as cardiovascular disease (https://academic.oup.com/eurjpc/article/28/14/1599/6145858), cancer (https://brieflands.com/articles/ijcm-110907.html, https://www.ncbi.nlm.nih.gov/pmc/articles/PMC8184167/), chronic kidney disease (https://journals.plos.org/plosone/article?id=10.1371/journal.pone.0254525#:~:text=The%20crude%20mortality%20rate%20among,19%20in%20the%20PIRP%20cohort.), among others. I found that the authors have not evaluated this factor. I suggest considering this factor in the analysis; if the data are not available, please address it in the limitations by using the mentioned references’

Response 3: Thank you for this comment and we have made the following change to our discussion and limitations:

‘Thirdly, whilst we considered the covariates to include in our analysis, we did not account for other co-morbidities such as cardiovascular disease,19 cancer20 and chronic kidney disease,21 all of which have been associated with worse outcomes in COVID-19.’ 

Reviewer #2

We thank the reviewer for his/her time and detailed, helpful comments and below are point-by-point responses to each of these. We have implemented the majority of the suggestions made by this reviewer, and provided explanations where we do not agree.

Comment 1: ‘Though the authors have improved the manuscript draft; There are still few confounding variables which could have impacted the outcome in second wave that needs to be evaluated and mentioned in the limitations. (e.g., CT score, day of presentation)’

Response 1: We thank the reviewer for this comment and have made the following changes to the discussion:

Fourthly, scoring systems based upon computer tomography (CT) have been used to characterize COVID-19 severity and outcomes (e.g. requirement of mechanical ventilation/mortality) in patients, particularly on presentation to hospital.22 CT scores were not included in this study as all patients were already intubated and ventilated. Furthermore, we are unable to comment if ventilatory data at time of intubation correlated with CT score as most patients had CT scans when clinically stable to undertake transfer to radiology departments, rather than specifically at the onset of mechanical ventilation.

Comment 2: ‘The authors mention "Although mortality risk in patients with COVID-19 is tempered by corticosteroids and tocilizumab" however fail to address this in the discussion.

Steroids – (50.9%) 1st wave vs (99.2%) of second wave

Tocilizumab –(6.3%) in 1st wave and (31.8%) of second wave

Remdesivir – (2.9%) in 1st wave and (41.6%) of second wave.?’

Response 2: We thank the reviewer for this comment and have made the following changes to the discussion:

We have added the following phrase to the discussion/limitations:

When we adjust for the key variables in our cox proportional risk analysis model, we find that in the second wave of the pandemic (despite use of repurposed therapeutics such as corticosteroids and Tocilizumab),13-15 there is no improvement in ICU mortality. This was particularly surprising considering the benefit of tocilizumab and corticosteroids in reducing COVID-19 mortality is well established.10,11

It may also be prudent to evaluate whether treatments found to have significant mortality benefit with older COVID-19 variants, continue to be successful as variants mutate and pathophysiology potentially alters. 

Comment 3: ‘The discussion still needs improvement based on the above facts.’

Response 3: Thank you for your comments regarding our discussion and conclusion. We agree and hope your suggested changes have imporeved the manuscript

---

## [Decision Letter · Decision Letter 2]

18 May 2022

Clinical and Survival differences during separate COVID-19 surges: Investigating the impact of the Sars-CoV-2 alpha variant in critical care patients

PONE-D-21-31797R2

Dear Dr. Soni,

We’re pleased to inform you that your manuscript has been judged scientifically suitable for publication and will be formally accepted for publication once it meets all outstanding technical requirements.

Kind regards,

Farzad Taghizadeh-Hesary

Academic Editor

PLOS ONE

Additional Editor Comments (optional):

Reviewers' comments:

Reviewer's Responses to Questions

**Comments to the Author**

1. If the authors have adequately addressed your comments raised in a previous round of review and you feel that this manuscript is now acceptable for publication, you may indicate that here to bypass the “Comments to the Author” section, enter your conflict of interest statement in the “Confidential to Editor” section, and submit your "Accept" recommendation.

Reviewer #3: (No Response)

2. Is the manuscript technically sound, and do the data support the conclusions?

Reviewer #3: Yes

3. Has the statistical analysis been performed appropriately and rigorously? 

Reviewer #3: Yes

4. Have the authors made all data underlying the findings in their manuscript fully available?

Reviewer #3: Yes

5. Is the manuscript presented in an intelligible fashion and written in standard English?

Reviewer #3: Yes

6. Review Comments to the Author

Reviewer #3: (No Response)

7. PLOS authors have the option to publish the peer review history of their article (what does this mean?). If published, this will include your full peer review and any attached files.

Reviewer #3: No

---

## [Editor Report · Acceptance letter]

23 Jun 2022

PONE-D-21-31797R2 

Clinical and Survival differences during separate COVID-19 surges: Investigating the impact of the Sars-CoV-2 alpha variant in critical care patients 

Dear Dr. Soni:

I'm pleased to inform you that your manuscript has been deemed suitable for publication in PLOS ONE. Congratulations! Your manuscript is now with our production department. 

Kind regards, 

on behalf of

Dr. Farzad Taghizadeh-Hesary 

Academic Editor

PLOS ONE